# Slowly-adapting type II afferents contribute to conscious touch sensation in humans: Evidence from single unit intraneural microstimulation

Roger Holmes Watkins[1] 🆔, Mario Durao de Carvalho Amante[2], Helena Backlund Wasling[2], Johan Wessberg[2] 🆔 and Rochelle Ackerley[1] 🆔

[1] *Aix Marseille Univ, CNRS, LNC (Laboratoire de Neurosciences Cognitives – UMR 7291), Marseille, France*
[2] *Department of Physiology, University of Gothenburg, Gothenburg, Sweden*

Edited by: Richard Carson & Vaughan Macefield

The peer review history is available in the Supporting information section of this article (https://doi.org/10.1113/JP282873#support-information-section).

**Abstract** Slowly-adapting type II (SA-II, Ruffini) mechanoreceptive afferents respond well to pressure and stretch, and are regularly encountered in human microneurography studies. Despite an understanding of SA-II response properties, their role in touch perception remains unclear. Specific roles of different myelinated A$\beta$ mechanoreceptive afferents in tactile perception have been revealed using single unit intraneural microstimulation (INMS), via microneurography, recording from and

**Roger Holmes Watkins** is a postdoctoral researcher, based at Aix-Marseille University, France. He obtained his PhD investigating the contribution of primary afferents to pathological pain states at the University of Bristol, UK. In 2014, he was awarded an MRC post-doctoral grant to learn microneurography at the University of Gothenburg, Sweden. He continues to record from and stimulate single afferents to directly investigate the relationship of afferent activity to touch in humans.

then electrically stimulating individual afferents. This method directly links single afferent artificial activation to perception, where INMS produces specific 'quantal' touch percepts associated with different mechanoreceptive afferent types. However, SA-II afferent stimulation has been ambiguous, producing inconsistent, vague sensations, or no clear percept. We physiologically characterized hundreds of individual A$\beta$ mechanoreceptive afferents in the glabrous hand skin and examined the subsequent percepts evoked by trains of low amplitude INMS current pulses ($<10\ \mu$A). We present 18 SA-II afferents where INMS resulted in a clear, electrically evoked sensation of large ($\sim$36 mm$^2$) diffuse pressure, which was projected precisely to their physiologically-defined receptive field in the skin. This sensation was felt as natural, distinctive from other afferents, and showed no indications of multi-afferent stimulation. Stimulus frequency modulated sensation intensity and even brief stimuli (4 pulses, 60 ms) were perceived. These results suggest that SA-II afferents contribute to perceived tactile sensations, can signal this rapidly and precisely, and are relevant and important for computational models of touch sensation and artificial prosthetic feedback.

(Received 14 February 2022; accepted after revision 3 May 2022; first published online 15 May 2022)

**Corresponding author** R. H. Watkins: Aix Marseille Univ, CNRS, LNC (Laboratoire de Neurosciences Cognitives–UMR 7291), 3 place Victor Hugo, Marseille 13003, France.    Email: roger.watkins@univ-amu.fr

**Abstract figure legend** Using microneurography, recordings were made from single mechanoreceptive afferents in the median nerve of human subjects. After fibre classification, low amplitude ($<10\ \mu$A) intraneural microstimulation was delivered to evoke sensations of touch. Varied sensations were evoked that could be attributed to selective activation of the recorded afferents. We identify a consistent link between type II slowly adapting mechanoreceptive afferents (SA-IIs) and a specific sensation (light pressure). These sensations matched the afferent properties precisely, indicated sensations were evoked by stimulating single SA-II afferents, and were modified by stimulus train modulations.

### Key points

- Slowly adapting type II mechanoreceptors (SA-IIs) are primary sensory neurons in humans that respond to pressure and stretch applied to the skin.
- To date, no specific conscious correlate of touch has been linked to SA-II activation.
- Using microneurography and intraneural microstimulation to stimulate single sensory neurons in human subjects, we find a specific sensation linked to the activation of single SA-II afferents.
- This sensation of touch was reported as gentle pressure and subjects could detect this with a high degree of accuracy.
- Methods of artificial tactile sensory feedback and computational models of touch should include SA-IIs as meaningful contributors to the conscious sensation of touch.

## Introduction

Precise tactile input from the glabrous skin of the hand is critical for the dexterous manipulation of objects and sensing our environment. Rich details about different aspects of these interactions are conveyed by our peripheral mechanoreceptive afferent system and are of importance to appropriately shape tactile behaviour. In the glabrous skin of the human hand, this information is encoded primarily by four types of cutaneous, fast-conducting, myelinated, A$\beta$ mechano-receptive afferent. These are (with their purported endings): fast-adapting type I (FA-I; Meissner corpuscles), fast-adapting type II (FA-II; Pacinian corpuscles), slowly-adapting type I (SA-I; Merkel complexes), and slowly-adapting type II (SA-II; Ruffini endings) afferents

(Vallbo & Johansson, 1984). Studies generally find that dynamic aspects of touch (vibration, fine texture) are conveyed by the fast adapting afferents and static aspects (edges, pressure, form) are conveyed by the slowly adapting afferents (Blake et al., 1997; Condon et al., 2014; Goodwin et al., 1997; Johansson & Birznieks, 2004; Johnson et al., 2000).

SA-IIs optimally encode tactile force, direction, and velocity, including fingertip grip forces that are important for interacting with objects (Birznieks et al., 2009; Westling & Johansson, 1987) and skin stretch/tension cues linked to proprioception (Aimonetti et al., 2007; Birznieks et al., 2001, 2009; Edin, 2001; Edin & Abbs, 1991; Grill & Hallett, 1995; Hulliger et al., 1979; Johansson, 1978; Johnson, 2001; Westling & Johansson, 1987). Additionally,

comparing psychophysical observations of directional skin stretch sensitivity to response characteristics of SA-II afferents in hairy skin suggests that SA-IIs in this skin region contribute to directional touch perception (Olausson et al., 1998). Historically, defining links between activity in particular afferent populations and aspects of tactile sensation has been based on correlating afferent activity recorded in non-human primates with measures of tactile perception in humans. SA-IIs have not been reported in non-human primate glabrous hand skin (Darian-Smith et al., 1980; Johnson, 2001), despite earlier reports (for an overview, see Knibestöl & Vallbo, 1970) and their clear presence in humans (Knibestöl & Vallbo, 1970; Vallbo & Johansson, 1984) and other mammals (Leem et al., 1993; Walcher et al., 2018). This has meant that the link between SA-II activity in non-human primates and perceived sensation has received little attention. Furthermore, non-human primate studies on glabrous hand skin have been used as a basis for the modelling of afferent contribution to human touch (Delhaye et al., 2019; Saal et al., 2017); hence, SA-II afferents have not been included in these.

However, in humans, no clear link has been demonstrated between SA-II afferent activation and a particular, singular touch sensation. This may be partly a result of the relative paucity of Ruffini end organ receptors in glabrous hand skin (Paré et al., 2003) and their lower incidence in single afferent microneurography studies (Johansson & Vallbo, 1979; Vallbo & Johansson, 1984), meaning that it is challenging to obtain sufficient population level data for these afferents. In addition to correlating mechanoreceptor firing and perception, it is possible to investigate links between mechanoreceptive afferent activation and tactile perception directly, via the technique of intraneural microstimulation (INMS) (Torebjörk & Ochoa, 1980; Torebjörk et al., 1987; Vallbo, 1981). Here, a single myelinated fibre can be selectively electrically stimulated during microneurography to evoke a 'quantal', isolated tactile percept.

In single unit INMS, an electrode is inserted into a human peripheral nerve and a single mechanoreceptive afferent is sought and characterized. Subsequently, low current ($<10$ $\mu$A) trains of INMS are delivered through the same electrode to activate the same afferent recorded from. This evokes an artificial tactile sensation projected to a specific location in the skin: the 'perceptive field'. Around half the time, there is a precise correspondence between the physiological receptive field and the electrically induced perceptive field (Sanchez Panchuelo et al., 2016; Torebjörk et al., 1987). When there is a match between receptive and perceptive properties, this suggests single afferent activation and links between activity in an afferent and tactile sensation can be explored. The percepts generated by different cutaneous A$\beta$ mechanoreceptive afferents can be readily described

by participants and have well-defined locations/qualities. Sensations linked to FA-I afferents are reported as focal vibration/tingle, sensations matching SA-Is are reported as focal pressure/internal pulling, and FA-IIs feel like a larger area of vibration (Macefield et al., 1990; O'Neill et al., 2019; Ochoa & Torebjörk, 1983; Sanchez Panchuelo et al., 2016; Schady & Torebjörk, 1983; Torebjörk & Ochoa, 1980; Trulsson et al., 2001; Vallbo, 1981; Vallbo et al., 1984).

Relatively few studies have used this approach because of its complicated and technically demanding nature, although the reports from INMS of SA-IIs have been inconsistent. Initial studies typically found no consistent sensation linked to SA-IIs (Ochoa & Torebjörk, 1983; Schady, Torebjörk et al., 1983; Torebjörk & Ochoa, 1980; Vallbo, 1981). The main consensus from subsequent work was that sensations may be linked to recordings from individual SA-IIs, but that these were 'strikingly non-uniform'. Studies by Schady and Torebjörk (1983) and Vallbo et al. (1984) did identify potential links to different types of sensations, including sustained lateral pulling of the skin, flutter, tapping, buzzing, vibration, pressure, and pain. Furthermore, Vallbo et al. (1984) found that the quality of the sensation ranged from feeling completely unnatural to almost natural in their four SA-II units; however, they noted sensations did have consistently large perceptive fields ($\sim$36 mm$^2$) with diffuse borders. In the following years, additional studies reported potential associations of pressure, swelling, joint movement, or strain (Kunesch et al., 1995; Macefield et al., 1990), but again these were not consistent and based on relatively few examples. By contrast to single unit INMS studies, which have correspondence between the receptive and perceptive fields, a study by Schady, Torebjörk et al. (1983) used INMS, without physiologically linking the percept. In this study, it was possible to elicit large areas of pressure sensation ($>$80 mm$^2$), which is much greater than what is found typically for SA-Is ($\sim$5 mm$^2$; Vallbo et al. 1984). Despite apparent contributions of SA-II afferents in the hairy skin to directional stretch perception, information on INMS when recording from these afferents has not been directly reported in the few studies examining INMS in hairy skin regions (Nagi et al., 2019; Schady & Torebjörk, 1983; Schady, Torebjörk et al., 1983). Overall, although the general opinion is that activating SA-IIs during INMS does not produce clear and consistent tactile sensations (Ochoa, 2010; Vallbo, 2018), this is based on somewhat conflicting evidence and there is scope for this to be explored and clarified.

We examined percepts linked to single A$\beta$ mechanoreceptive afferents in the glabrous skin of the hand using INMS. We aimed to characterize the INMS evoked sensations linked to individual characterized SA-II afferents, comparing these with sensations generated by

stimulating the physiologically similar SA-Is, which have a well-defined associated percept of focal pressure.

## Methods

### Ethical approval

The studies were approved by the local University of Gothenburg (628-17) and University of Nottingham (E09022012) ethics committees. Healthy adult human participants received information about the study before participating and signed written informed consent forms. The experiments were performed in accordance with the *Declaration of Helsinki*, except for registration in a database.

### Experimental approach

The data were collected over three different experimental series, each having different aims, although the over-arching aim of all studies was to perform single unit INMS to examine evoked sensations linked to individual myelinated A$\beta$ mechanoreceptive afferents. All studies used microneurography to record from single afferents in the median nerve at the wrist and subsequent INMS. The first experimental series (A) was conducted at the University of Gothenburg, Sweden, and its aim was to quantify perceptual limits of frequency discrimination using single unit INMS. The second experimental series (B) combined single unit INMS with concurrent neuroimaging at the University of Nottingham, UK (O'Neill et al., 2019; Sanchez Panchuelo et al., 2016) and investigated the patterns of cortical activity induced by INMS evoking single afferent linked tactile sensations. The third experimental series (C) was conducted at the University of Gothenburg, Sweden, and aimed to characterize the temporal integration of INMS evoked single afferent tactile sensations.

In total, 12 participants with stimulated SA-II units were included in the present work, of which seven were in experimental series A (mean ± SD: 24 ± 2 years, five females), two in series B (39 ± 6 years, two females), and three in series C (30 ± 8 years, one female). At the beginning of experimental series C, it was clear from the results obtained in experimental series A and B that, in contrast to previous studies, distinctive, consistent and specific sensations were associated with physiologically-identified SA-II afferents. Therefore, this experimental series was considered to have the least methodological bias against stimulation of SA-II afferents because, previously, these would have been assumed to not be successfully activated by INMS and were not always tested. In experimental series C, all single afferents encountered, including all SA-IIs, were thoroughly examined and characterized, and subsequently tested with INMS. Therefore, for specific comparisons on proportions, all the A$\beta$ mechanoreceptive afferent units that were microstimulated in series C are compared, involving 13 participants (27 ± 5 years, seven females; two participants were also included in the group in series C above).

Single unit recordings were made from myelinated A$\beta$ mechanoreceptive afferents in human participants using microneurography according to standard procedures (Vallbo & Hagbarth, 1968) and the procedure for performing the single unit INMS is outlined in Fig. 1. The median nerve, which projects to the majority of the glabrous skin of the hand, was accessed ∼3 cm proximal to the wrist; for more details, see Sanchez Panchuelo et al. (2016) and O'Neill et al. (2019) from experimental series B. Commercial tungsten micro-electrodes (model UNA15FNM; FHC, Bowdoin, ME, USA) with tip diameters of ∼5 $\mu$m were inserted perpendicularly into the skin and adjusted to an intraneural position for neural recordings. Electrodes were connected to custom designed amplifiers capable of amplifying and bandpass filtering nerve activity (∼0.2–4 kHz) for display and saving. Experimental series A used the SC/Zoom setup (University of Umeå, Umeå, Sweden) and experimental series B and C used the micro-neurography recording/stimulation setup detailed in Glover et al. (2017). Both systems had integrated nerve stimulators, which were capable of delivering stimulation at intensities of up to ∼200 $\mu$A with a ≤0.1 $\mu$A resolution. Up to 10 $\mu$A of stimulation was used in the present experiments, delivering intraneural stimulation through high impedance electrodes. Using these electrodes, matched recording and stimulation in single afferents (see below) was achieved with electrodes having impedances typically measured as between 150 and 500 kΩ *in situ* at 1 kHz (Glover et al., 2017). Single unit recordings were identified using manual stimulation of the participants' hand skin to evoke mechanical responses. Once isolated, units were classified as FA-I, FA-II, SA-I, or SA-II based on standard criteria (Vallbo & Johansson, 1984).

For the specific identification and separation of slowly-adapting mechanoreceptive afferents, several criteria may be used (Chambers et al., 1972; Gynther et al., 1992; Johansson, 1978; Knibestöl, 1975; Vallbo & Johansson, 1984; Wellnitz et al., 2010). We distinguished SA-IIs from SA-Is by their sensitivity to remote skin stretch (SA-IIs are sensitive, but SA-Is are not), whether they were spontaneously active (often seen in glabrous skin SA-IIs, but not SA-Is), and their firing properties during sustained indentation (Fig. 1*A*). During sustained indentation, SA-IIs show lower initial dynamic sensitivity than SA-Is and high regularity in their firing during the static phase of adaptation, with a coefficient of variation typically of <0.3 compared to >0.5 for SA-Is. This

difference in adaptation was qualitatively assessed in all SA-IIs from experimental series A using the coefficient of variation measure detailed below, and was qualitatively assessed online in both SA-I and SA-II afferents in series B and C. This measure provided the clearest differentiation of the SA-I and SA-II afferents, but an irregularity index measure that takes into account slow drifts in responses could additionally be employed as a more sensitive method to differentiate the slowly adapting afferents, as utilized in analysis of regularity in muscle afferent firing patterns (Birznieks et al., 2008).

After the initial functional classification, liminal stimuli with calibrated monofilaments were used to identify the most sensitive region in the receptive field (usually a single spot for SA-II afferents and a small region of points of high sensitivity for SA-I afferents; Johansson, 1976), which was used as the point for testing the correspondence

with perceptual sensations (see below). In experimental series A for an analysis of firing regularity, a sustained indentation was delivered at this point with a supra-threshold monofilament for a period of ~30 s. After the initial burst firing on application of the monofilament, the coefficient of variation of firing in the static phase of the response to a controlled sustained indentation was calculated to gain the standard deviation of firing over at least 100 interspike intervals (Chambers et al., 1972; Gynther et al., 1992; Knibestöl, 1975). Furthermore, SA-IIs in glabrous skin typically have larger receptive fields than SA-Is (~59 mm$^2$ compared to ~11 mm$^2$, respectively; Vallbo & Johansson 1984), although this is not necessarily a reliable classifier of afferent type (Knibestöl & Vallbo, 1970). In line with the aims of series B and C, the receptive field extent was additionally more accurately assessed by suprathreshold calibrated

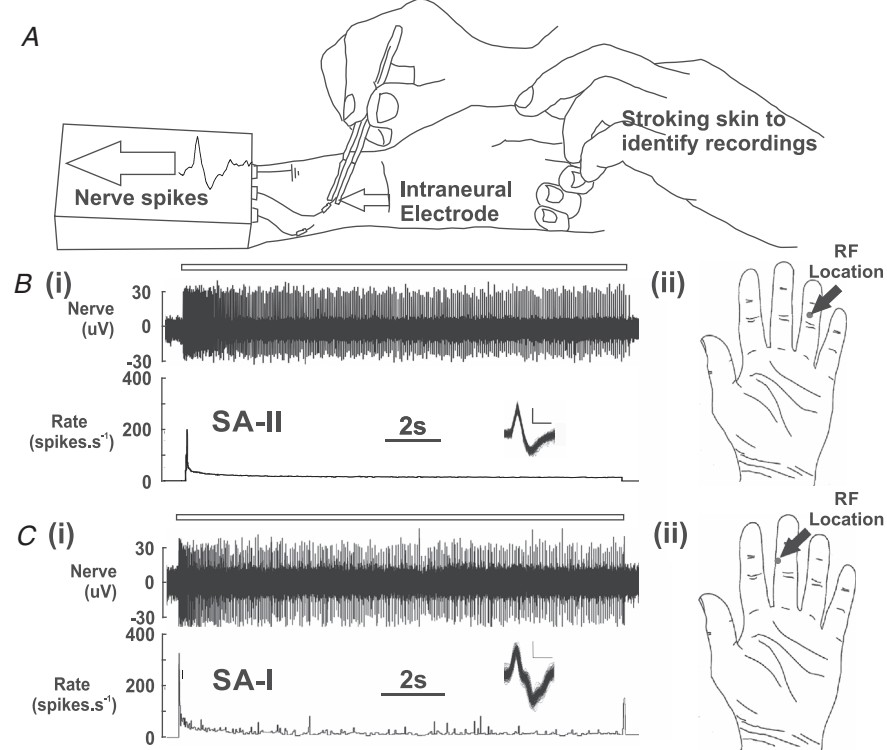

**Figure 1. Methods for single unit microneurography recordings**
*A*, microneurography recording procedure used to identify single mechanoreceptive afferents in the glabrous skin of the hand. *A*, experimental set up for recording showing the afferent identification procedure. *B*, characterization of an SA-II afferent (Fig. 3, unit A6) through (i) physiological classification using sustained indentation (calibrated monofilament of 44 mN) and (ii) a schematic representation of the identified receptive field (RF) location on the hand. This afferent showed very little variation in its firing, as seen in the nerve trace and firing rate, and had a coefficient of variation of 0.14 in the static phase of the response to sustained indentation. *C*, characterization of an SA-I afferent (Fig. 4, unit C12) through (i) physiological classification using sustained indentation (calibrated monofilament of 30 mN) and (ii) a schematic representation of the identified receptive field (RF) location on the hand. This afferent showed much more variation in its firing, as seen in the nerve trace and firing rate, and had a coefficient of variation of 0.53 in the static phase of adaptation. For both afferents, the bar above the trace indicates the timing of mechanical stimulation and inset traces display overlaid spikes for the entire response (scale bar = 0.5 ms, 20 $\mu$V).

monofilaments (∼four times the activation threshold) around the most sensitive point. Using this method, all measured SA-I afferents had receptive field diameters of ≤3 mm (<7 mm$^2$), whereas all measured SA-II receptive field diameters were ≥7 mm (>40 mm$^2$), similar to previous studies (Vallbo & Johansson 1984). In all the afferents reported in the present study, SA-I and SA-II afferents were clearly and unequivocally distinguished on the basis of their response characteristics as defined prior to commencing INMS.

After isolating and characterizing the individual afferents, INMS was delivered via the recording electrode using trains of positive 200 μs current pulses. In series B and C, the pulses were followed by a full active charge-balancing, using a 2 ms negative pulse (Glover et al., 2017). To initially identify and characterize sensations, afferents were stimulated using pulse trains of 30 or 60 Hz, for 500 or 1000 ms, depending on the experiment. Systematic comparisons of frequency (50–300 Hz) were performed in series C, but various parameters (including other frequencies between 15 and 600 Hz, at 500 or 1000 ms duration) were non-systematically tested in series A and B during the test paradigm and impacts on sensation were noted. For each afferent encountered and characterized, INMS pulse trains were repetitively delivered as the current intensity was gradually increased from 0 μA in increments of 0.1 μA or less, repetitively delivering pulse trains, until participants reported a singular clear tactile sensation in the hand (typically ∼1−3 μA), or up until a maximum of 10 μA. This generated sensation was termed the 'perceptive field' (Vallbo et al., 1984) (Fig. 2A). The intensity of stimulation was then slightly increased to ensure a clear sensation, which was taken to indicate a successful 1:1 relationship between current pulses and afferent activation. Whenever the current was increased and the sensation changed (usually the recruitment of a second discrete percept or a larger sensation of paraesthesia), no further stimulation tests were performed because this is considered to be multi-unit activation (Vallbo et al., 1984). The stimulation intensity was thus set at a slightly suprathreshold intensity around this level to maintain sensation integrity and reduce the possibility for non-specific sensations. The sensations generated by reference stimuli (e.g. 60 Hz for 1 s) were monitored throughout to control for changes in electrode position/threshold.

The correspondence between the receptive field of the individual identified afferent and the INMS evoked perceptive field was verified using alternating mechanical and electrical stimulation. Localized mechanical stimulation was delivered using indentation with a blunt wooden stick (tip diameter ∼1 mm) at the identified point of maximal sensitivity alternately with the pulse trains used for initial identification of sensations. Stimulation was only continued if: (1) there

was a very close correspondence between the physiological receptive field and the INMS evoked receptive field, with exactly overlapping percepts or located within a few mm (Torebjörk et al. 1987; Vallbo et al. 1984) (Fig. 2B) and (2) if the quality of the sensation (cyclic vs. sustained) matched the adaptation characteristics of the afferent (fast vs. slow adaptation, respectively). This careful evaluation of correspondence between stimulation and recording generally results in a ratio of successful single unit INMS of ∼50% following recordings, where the sensation is isolated and linked to the physiologically characterized afferent (Sanchez Panchuelo et al., 2016; Torebjörk et al., 1987). In experimental series C, all characterized single afferents were noted, as well as the number of successfully matching percepts; importantly, in experimental series C, this included testing of all SA-II afferents. In experimental series A and B, this was not tested systematically.

After the initial identification, the participant evaluated the sensation induced from INMS, receiving the visual prompts (Fig. 3C–E), including the perceived sensation size [0.01 (very small point), 0.1 (small point), 1, 2, 3, 4,

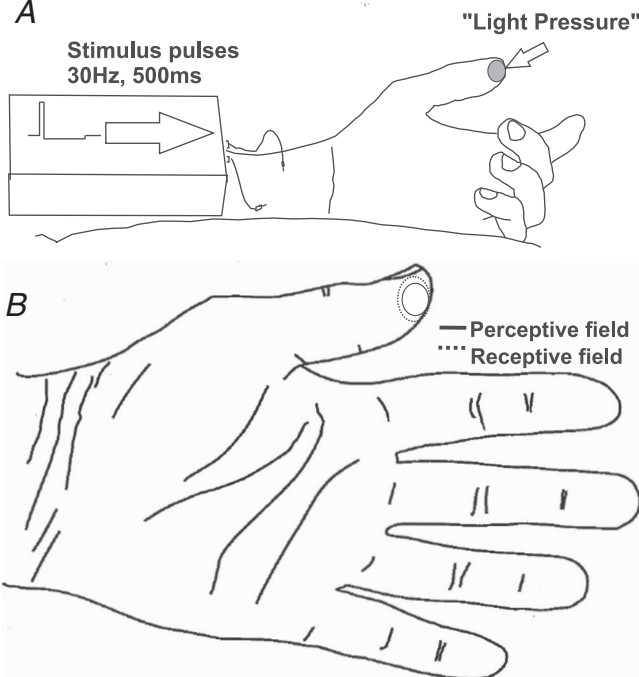

**Figure 2. Intraneural microstimulation (INMS) procedure**
A, localization and description of the initial INMS-evoked percepts from stimulating a single afferent. An oval region of light pressure was evoked by 500 ms of 30 Hz stimulation (Fig. 3, unit C2). B, close correspondence of the physiological receptive field and evoked perceptive field in the same afferent (unit C2). The perceptive field was oval and 10 mm in diameter and overlapped with the most sensitive region in the receptive field (hot spot). The full extent of the receptive field is also illustrated, which was oval and ∼12 mm in diameter, as mapped by suprathreshold monofilament stimulation.

5, 6, 7, 8, 9, 10, or >10 mm diameter], border (1 = sharp, 2 = slightly diffuse, 3 = clearly diffuse, 4 = single area with points of intense sensation), shape (1 = round, 2 = oval, 3 = long, 4 = irregular but continuous, 5 = many separate close points), whether there was any sensation of movement across the skin (1 = no movement, 2 = linear movement, 3 = circular movement), and how natural it felt (1 = completely natural, 2 = almost natural, 3 = possibly natural, 4 = rather unnatural, 5 = completely unnatural; not shown) Vallbo et al. (1984). Participants were also able to give half numbers (e.g. 2.5 for a perceptive field size between 2−3 mm diameter) where applicable. Additionally, participants were asked to provide a qualitative description of the evoked sensation in their own words, which was noted.

In experimental series C, we systematically tested the minimum duration of stimulation required to perceive a sensation and how changing the frequency of stimulation modulated the sensation. To assess the minimum duration of pulses that could reliably be perceived, participants were asked to report the presence of a detectable sensation associated with a visual cue. Participants received a visual cue signalling the brief stimulation, with no cue as to its temporal duration (word 'STIM' displayed on the screen

for a duration of 1 s. The stimulation protocol began at 50 Hz for 400 ms (21 pulses), where sensations were reliably perceived in all trials. The stimulation duration was decreased in steps from 300 ms (16 pulses), then 200 ms (11 pulses), 80 ms (5 pulses), and down to 40 ms (3 pulses), until the sensation was no longer perceived or the minimum duration of 40 ms was reached. At this end point, the stimulus duration was increased again until the sensation was re-established. This procedure was repeated a few times around these values to give a threshold for the minimum duration of stimulation generating a percept. This was followed by paired presentations of different frequencies from 50 to 300 Hz (400 ms duration at 50/75/100/150/200/250/300 Hz). Participants were required to say if there was a discernible difference in the sensation after each paired stimulus presentation, with a report on how it differed.

## Results

In total, 18 SA-II afferents were characterized physiologically and were found to have electrically-evoked perceptive fields that matched the physiologically-defined

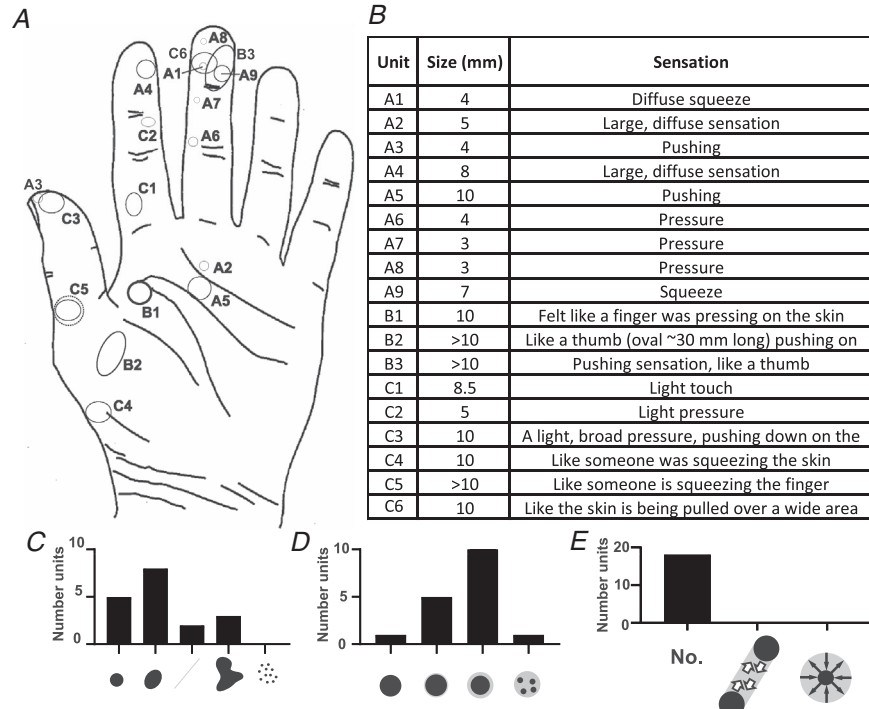

**Figure 3. SA-II evoked perceptive field properties from single unit intraneural microstimulation (INMS)**
*A*, size and location of electrically-evoked perceptive fields associated with single SA-II afferents. Approximate receptive field sizes are scaled to the hand, a single unit (C5) had a perceptive field of >10 mm diameter and is indicated with a dashed outline, but was not mapped as for the other units of >10 mm (B2/B4). *B*, qualitative descriptions of sensations associated with SA-II afferents, unit codes correspond to those indicated on the hand map in (*A*). *C* to *E*, quantitative characterization of SA-II linked percepts for perceptive field size (*C*), its border (*D*), and if there was any perceived movement (*E*).

receptive fields (Fig. 3). Two of the SA-II units were spontaneously active (units A3 and A9). The mean threshold current required for sensation perception was 3.4 $\mu$A (range 1.2–8.6 $\mu$A). All SA-II linked percepts were recruited in an all-or-none manner, with a defined current threshold for the first sensation and invariant quality/intensity when increasing above this (Vallbo et al. 1984), similar to all other mechanoreceptive afferents tested in our three experimental series (O'Neill et al., 2019; Sanchez Panchuelo et al., 2016). The qualitative descriptions of the evoked sensation from SA-IIs were of a single region of pressure, light touch, or squeeze. A number of participants exclaimed how realistic the sensation felt, where they spontaneously described the touch as if someone else's digit was pushing down on their skin (Fig. 2*B*).

The locations and relative sizes of the SA-II perceptive fields are illustrated in Fig. 3*A*. These perceptive fields were relatively large (median diameter = 8 mm, range = 3−30 mm; Fig. 3*B*), corresponding to approximate areas of 10−700 mm$^2$. The sensation perceived was typically elongated in shape (median = oval, range = round to irregular; Fig. 3*C*) with a diffuse border (median = clearly diffuse, range = defined to clearly diffuse; Fig. 3*D*), consisting of a single/continuous area of sensation that was not accompanied by any sensation of movement across the skin (Fig. 3*E*), and was generally described as natural (median = completely-almost natural, range = completely natural to rather unnatural). The qualitative descriptions of the evoked sensation were of a single region of pressure, light touch, or squeeze (Fig. 3*B*). A number of participants independently exclaimed how realistic the sensation felt, where they described the touch as if someone else's digit was pushing down on their skin in time with the duration of the stimulation (Fig. 3*B*).

### Effect of modifying stimulus frequency and duration on sensation perception

In 11 units over the three experimental series, sensations resulting from stimulation at different frequencies revealed an invariance in the quality of the perceived pressure/squeezing sensation from SA-IIs. In general, as the frequency of stimulation increased, the perceived pressure/squeezing sensation became more intense. It is noteworthy that increasing the stimulation frequency changed the sensation in two different SA-II units located on the middle fingertip (Fig. 3, units A9 and C6), where, in addition to increased pressure on the skin, it began to feel like tension applied to the nail. In eight SA-IIs in experimental series C, systematically increasing the stimulation frequency monotonically increased the intensity of the percept (stronger pressure squeeze).

Five of these SA-II units were tested further, where stimulation frequencies over 150 Hz (up to 300 Hz) did not increase the perceived intensity. The minimum consistently perceived duration of stimulation was also tested. In five SA-IIs that were tested with decreasing stimulus durations at 50 Hz frequency, 4 pulses (duration 60 ms) reliably produced a perceivable sensation in four units and 11 impulses (duration 200 ms) was needed to reliably produce a percept in the other unit.

We compared percepts generated from stimulating the other type of slowly-adapting units, that is SA-Is, from experimental series B and C. This group of SA-Is ($n = 24$) produced clearly different sensations from the SA-II afferents, where SA-I stimulation evoked predominantly point-like perceptive fields (mostly <1 mm diameter; Fig. 4). These had well-defined borders and the sensations were typically described as a sharp pencil indentation, pinching, or internal pulling (Fig. 4*C*). On comparing these SA-I percepts with the full group of 18 SA-IIs, there was a small overlap in perceptive field size (at ∼3−5 mm diameter), but the sensations were always qualitatively distinct (Figs 3*B* and 4*C*).

We examined the incidence of matching physiological receptive fields with the perceptive field in experimental series C, where all characterized units were noted and tested. Out of a total of 103 single A$\beta$ mechanoreceptive afferents in series C, 44% (46/104) were linked to a perceptive field that matched precisely the physiological receptive field. Concerning SA-IIs, 43% (6/14) had matching sensations compared to 44% (40/90 units) for all the other A$\beta$ mechanoreceptive afferents, suggesting no significant difference in the effectiveness of evoking matching sensations after recording from SA-II afferents.

### Discussion

In the present study, we have defined a specific, consistent percept associated with electrical stimulation of SA-II afferents in the glabrous skin of the hand, using the technique of single unit INMS during microneurography. This is somewhat in contrast to early INMS studies, which generally found no consistent sensation (Ochoa & Torebjörk, 1983; Schady, Torebjörk et al., 1983; Torebjörk & Ochoa, 1980; Vallbo, 1981), although subsequent studies suggested a tentative link to large area sensations including pressure/strain, but this was based on a small number of observations (Kunesch et al., 1995; Macefield et al., 1990; Vallbo et al., 1984). From our sample of 18 SA-IIs, we found that a broad pressure/pushing/squeeze sensation was consistently associated with these afferents. This is clearly qualitatively different to stimulating fast-adapting mechanoreceptive afferents in glabrous skin that give cyclic percepts of vibration/buzzing, including flutter, tingle, and tapping sensations (Bini et al., 1984;

Macefield et al., 1990; O'Neill et al., 2019; Ochoa & Torebjörk, 1983; Sanchez Panchuelo et al., 2016; Trulsson et al., 2001; Vallbo et al., 1984), with evidence of similar sensations produced in hairy skin (Nagi et al., 2019; Ochoa & Torebjörk, 1983; Schady & Torebjörk, 1983).

## SA-II percepts, differences with SA-Is, and comparison with previous studies

Slowly-adapting mechanoreceptive afferents have been linked to *sustained* percepts of pressure when stimulated with INMS, which relates well to their encoding of continued indentation (Vallbo & Johansson, 1984). This has been well-studied in SA-I afferents in the glabrous skin, which typically produce a small, well-defined, point-like sensation of pushing/pulling (Macefield et al., 1990; Ochoa & Torebjörk, 1983; Sanchez Panchuelo et al., 2016; Schady & Torebjörk, 1983; Torebjörk & Ochoa, 1980; Trulsson et al., 2001; Vallbo, 1981), which we also found presently (Fig. 3). Our SA-II afferents also produced sustained 'quantal' percepts of pressure, but these were different to the SA-Is, with a larger perceptive field (Figs 2 and 3, SA-II median diameter = 8 mm compared to <1 mm in SA-Is) and much more diffuse

border. Additionally, the SA-II linked sensations were qualitatively different, described as a large skin squeeze, compared to SA-Is that felt like a small point of pressure/pulling. A previous study using INMS without recording occasionally evoked diffuse pressure percepts with a diameter >10 mm (Schady, Torebjörk et al., 1983), similar to our present findings, but did not link this to a specific afferent type.

One question our findings raise is why these sensations have not been well-documented and linked to consistent sensations previously. In the first single unit INMS studies conducted in the early 1980s, clear associations were not found between SA-II afferent recordings and percepts evoked by INMS, unlike in other Aβ mechanoreceptive afferents in the glabrous skin (Ochoa & Torebjörk, 1983; Schady, Torebjörk et al., 1983; Torebjörk & Ochoa, 1980; Vallbo, 1981). Some of these studies did identify potential percepts linked to SA-IIs, but these were highly variable. However, these potentially associated sensations did consistently have large perceptive fields (>30 mm²) with diffuse borders (Schady & Torebjörk, 1983; Schady, Torebjörk et al., 1983; Vallbo, 1981). This hints at a qualitative difference found when stimulating SA-II afferents, which appear to have a substantially larger area

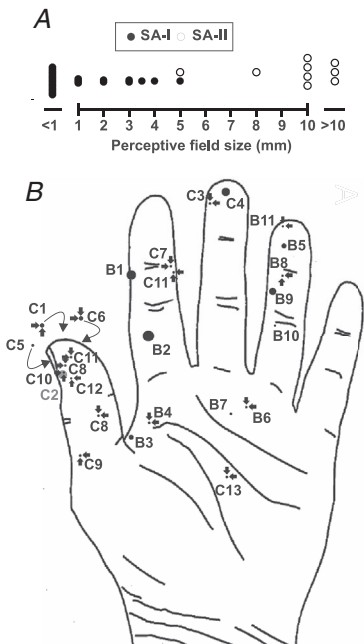

| Unit | Size (mm) | Sensation |
|------|-----------|-----------|
| B1 | 4 | Pinching |
| B2 | 5 | Pinching |
| B3 | 2 | Pulling |
| B4 | 0.1 | Like a pen pushing/internal pulsing |
| B5 | 2 | Tight pulling/tugging sensation |
| B6 | 0.1 | Pulling |
| B7 | 1 | Sharp, almost painful, pulling sensation |
| B8 | 0.1 | Point-like |
| B9 | 3 | Pressure |
| B10 | 1 | NA |
| B11 | 0.1 | Small point of pulling |
| C1 | 0.01 | Very small point of pressure |
| C2 | 3 | Point, sharp but not painful |
| C3 | 0.1 | Sharp point on the skin, no painful |
| C4 | 3/4 | Like someone is pressing a pencil |
| C5 | 1 | Pressing down, blunt pencil |
| C6 | 0.01 | Very small point, like a blunt needle |
| C7 | 0.1 | Like a tiny bug under the skin biting |
| C8 | 0.1 | Pressure, like someone is poking |
| C9 | 0.01 | Thin pencil push |
| C10 | 0.1 | Pencil pushing |
| C11 | 0.1 | Thin pencil push |
| C12 | 0.1 | Small pencil push |
| C13 | 0.1 | Needle pressing on the skin without pain |

**Figure 4. Perceptual correlates of single SA-I afferent linked percepts**
*A*, comparison of perceptive field diameters in SA-I (filled circles) and SA-II (close circles) afferent linked percepts. *B*, size and location of electrically-evoked perceptive fields associated with SA-I afferents. Approximate receptive field sizes are scaled to the hand, and afferents with perceptive fields <1 mm (0.1/0.01 mm, corresponding to a point/very small point, respectively) are indicated with arrows pointing to the perceptive field location. A single overlapping unit on the thumb is illustrated in grey, and locations of units on the dorsal side of the thumb are indicated within large arrows. A single SA-II linked perceptive field >10 mm diameter that was not fully mapped is indicated with a dashed outline (unit C5). *C*, qualitative descriptions of sensations associated with SA-II afferents, unit codes correspond to those indicated on the hand map in (*A*).

sensation than in other mechanoreceptive A$\beta$ afferent types. Subsequent work corroborated these qualitatively different and more diffuse percepts, where Macefield et al. (1990) found that 4/18 stimulated SA-IIs produced a perception that was of pressure, swelling, or the perception of joint movement and Kunesch et al. (1995) found a few SA-IIs that gave a matching sensation of strain. This is again distinct from the other single afferent linked sensations and also different from sensations generated when stimulating multiple afferents simultaneously, which is typically of an electric shock/paraesthesia (Schady, Ochoa et al., 1983), or when only a couple of nerve fibres are stimulated, sometimes generates separate small points (Schady, Torebjörk et al., 1983) or a line (Sanchez Panchuelo et al., 2016).

Thus, previous evidence was somewhat ambiguous, with some studies suggesting the possibility of a particular percept linked to SA-IIs. In our large sample of INMS evoked percepts that were linked to individual recorded afferents ($n = 250$), there was a clear association of a distinctive broad pressure percept with SA-II afferents. This was reliable across our three experimental series, where participants could clearly discern and describe a similar percept in all cases, and this was different compared to all other percepts associated with other single afferents using INMS.

## Methodological considerations in light of differences from previous work and future directions

It is beyond the scope of the present study to investigate reasons why SA-II linked sensations have not been consistently observed previously, although there are several plausible possibilities. A general problem when delivering INMS, and ascribing percepts to the activation of individual afferents, is the need for a precise correspondence between the physiological receptive field and perceptive field sensation. It is therefore possible to corroborate a perceived sensation and its afferent basis, but a negative finding is harder to address (i.e. the lack of a percept) because the effectiveness of stimulation cannot be established. An indirect method has been used to infer successful activation despite a lack of sensation (Torebjörk & Ochoa, 1980), but involves prolonged high frequency stimulation, which may induce significant nerve polarization. A direct, but more technically complicated, method can be used to examine afferent recruitment in INMS independently of subject reports of sensations; two electrodes are inserted into different proximal/distal locations of the same nerve and a paired fascicular position can be achieved (Ochoa & Torebjörk, 1983). In this case, a particular percept can be attributed to the recruitment of a unitary potential upon stimulation, which can be directly recorded using triggered averaging. However, the use of this technique is much more difficult than single unit INMS alone, which is already a challenging approach. Thus, this is not currently possible to use in routine INMS studies. However, future studies, assisted by technical developments such as ultrasound guidance (Dunham et al., 2018) and multicontact electrode recordings (Sales et al., 2022) may increase the practical feasibility of using this paired recording technique to investigate afferent recruitment and dependence on stimulation parameters directly.

In the present study, we performed INMS using charge balanced constant current stimulation, and commercial electrodes. This is in contrast to early studies that used self-fabricated electrodes (Vallbo 1981; Vallbo et al. 1984) and non-charge balanced constant voltage stimulation (Ochoa & Torebjörk 1983; Schady, Torebjörk et al., 1983; Torebjörk & Ochoa 1980). Thus, an intriguing possibility for the differences is that some aspect of this change of equipment used may have differed between studies because the factors determining successful INMS are not well understood (Torebjörk et al., 1987; Vallbo et al., 1984). Recruitment properties of afferents by electrical stimulation around threshold (as employed in the present study to avoid partial or non-specific afferent recruitment) are complex (Bostock, 1983; Bostock & Rothwell, 1997; Burke et al., 1998; Mogyoros et al., 1996). These may additionally differ between different types of peripheral axon (sensory *vs.* motor; Mogyoros et al., 1996); thus, there is scope to investigate these parameters on a single afferent level, aiming to better understand recruitment by electrical stimulation. This may be addressed directly in future studies changing stimulus parameters (pulse width/repolarization) or looking for associations between electrode characteristics (e.g. tip impedance) and success in evoking sensations. So far, studies have used trains of standardized 0.2–0.25 ms positive voltage/current pulses to recruit sensations. However, pulse width modulation clearly has effects on evoked sensations independently of amplitude, and the specific modulation of this may have an impact on recruited afferents and the electrically evoked sensations and their natural quality (Tan et al., 2014), potentially suggesting complex population effects of pulse configuration on afferent recruitment. A better understanding of the factors that determine afferent recruitment when stimulating electrically could have implications for the successful recruitment of SA-II afferents and if there are optimal parameters with which to stimulate these. This understanding could allow more successful generation of pressure sensation using stimulation methods such as juxtaneural or intraneural peripheral nerve stimulation in amputees to allow a more diverse range of available feedback (Ackerley et al., 2018; Oddo et al., 2016; Tan et al., 2014).

Our results suggest that SA-II afferents contribute a unique aspect to touch sensation, signalling the

overall pressure/strain experienced during tactile interactions. Estimating grip forces is critical in dexterous object manipulation, and sensing the overall pressure of these interactions allows us not only to avoid applying excessive pressure, but also to operate within the safety margin of grip force, below which microslips may occur (Delhaye et al., 2021; Khamis et al., 2014; Westling & Johansson, 1987). These microslips can be used for precise, extremely rapid, adjustment of grip, but only when maintaining forces extremely close to the point of object slippage. In terms of quasi-static tactile interactions, where fast-adapting mechanoreceptive afferents give minimal input, slowly-adapting mechanoreceptive afferents give detailed input on sustained mechanical skin deflection (Vallbo & Johansson, 1984). Comparing the sustained tactile percepts that we obtained using INMS, it is evident that both SA-I and SA-II afferents convey sensation related to tactile pressure, but do so in a complementary manner. The very focal pressure sensation evoked by SA-I afferent stimulation indicates the perceptual coding of very precise and local tissue indentation, whereas SA-II afferents give a broad sensation of pressure/strain. This suggests that, in object interactions, surface details are signalled by the SA-I afferents, whereas SA-IIs give information on the overall contact force and skin stretch, in line with the literature on their response properties (Aimonetti et al., 2007; Birznieks et al., 2001, 2009; Blake et al., 1997; Connor et al., 1990; Yoshioka et al., 2001). It is also noteworthy that brief durations of stimulation (3 pulses, 50 Hz) were perceived in SA-II microstimulation, albeit weakly, suggesting a role in the rapid and precise signalling of touch.

Despite identifying a direct link between SA-II afferents and tactile sensation in the glabrous skin of the hand, several questions still remain. The utilization of skin stretch cues in proprioceptive judgements, which will be optimally encoded in SA-II firing proprioception (Aimonetti et al., 2007; Birznieks et al., 2001, 2009; Edin & Abbs, 1991; Grill & Hallett, 1995; Hulliger et al., 1979; Johansson, 1978; Westling & Johansson, 1987), indicates an SA-II contribution to proprioception. However, proprioceptive sensations were not observed in the present study. This suggests that either summed activation in many SA-IIs is responsible for these sensations or that there is some kind of population coding based on differential directional sensitivities that contributes to this. Additionally, a directional pulling sensation was observed in only a single case, despite many SA-IIs possessing directional sensitivity (Edin, 2004; Vallbo & Johansson, 1984). Further investigations are required to identify whether any directional percepts are linked to particular afferent properties, although it is possible that information on directional skin stretch is conveyed in the population response of the SA-IIs. Furthermore, SA-II afferents in hairy skin have sub-stantially different response properties compared to glabrous skin SA-IIs (with much smaller receptive fields; Vallbo et al. 1995) and have been clearly implicated in the sensitive signalling of lateral stretch detection in hairy skin (Olausson et al., 1998). Thus, it would be of interest to compare the relative perceptual contributions of SA-II afferents in different skin areas.

## Conclusions

In conclusion, we have identified a specific perceptual tactile correlate of SA-II activation. Future work should examine the relationship between evoked SA-II activity and aspects of somatosensory perception, as well as the potential for subtypes of SA-IIs (e.g. nail SA-IIs, differences between glabrous and hairy skin), both physiologically and through INMS, which highlights the importance of thoroughly characterizing peripheral afferents. In the present study, INMS consisted of trains of stimulation at a constant frequency, but it is of interest to quantify the relationship of the evoked sensations with comparative physiological stimuli (indentation/velocity) mimicking the dynamic firing patterns to tactile stimuli, and how this may shape the perception.

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

## Additional information

### Data availability statement

All data are presented in the manuscript and figures.

### Competing interests

The authors declare that they have no competing interests.

## Author contributions

The experiments were performed at the Department of Physiology at the University of Gothenburg, Sweden and at the Sir Peter Mansfield Imaging Centre at the University of Nottingham, UK. RW and RA were responsible for the conception of work. RW, MA, HBW, JW and RA were responsible for the acquisition of data for the work. RW and RA were responsible for the analysis of data for the work. RW, MA, HBW, JW and RA were responsible for the interpretation of data for the work. RW and RA were responsible for drafting the work. MA, HBW and JW were responsible for critically revising the work for important intellectual content. All of the authors approved the final version of the manuscript submitted for publication; agree to be accountable for all aspects of the work in ensuring that questions related to the accuracy or integrity of any part of the work are appropriately investigated and resolved; and confirm that all persons designated as authors qualify for authorship, and all those who qualify for authorship are listed.

## Funding

This work was supported by the Swedish Research Council (grant 2017-01717 to JW), Sahlgrenska University Hospital (ALFGBG grant 725751 to JW), a European Research Council (ERC) Consolidator grant under the European Union's Horizon 2020 research and innovation programme (grant agreement No. 772242 to RA), and the UK Medical Research Council (TOUCHMAP, grant number MR/M022722/1).

## Acknowledgements

We thank Karin Göthner for technical assistance at the University of Gothenburg. We also thank Susan Francis, Paul Glover, Rosa Sanchez Panchuelo, and the rest of the TOUCHMAP team at the Sir Peter Mansfield Imaging Centre at the University of Nottingham for their experimental support in experimental series B.

## Keywords

mechanoreceptor, pressure, Ruffini, SA-II, skin, tactile, touch

## Supporting information

Additional supporting information can be found online in the Supporting Information section at the end of the HTML view of the article. Supporting information files available:

**Statistical Summary Document**
**Peer Review History**

