## [Peer Review History · The Journal of Physiology]

Slowly-adapting type II afferents contribute to conscious touch sensation in humans: evidence from single unit intraneural microstimulation

Roger Holmes Watkins, Mário Durão De Carvalho Amante, Helena Backlund Wasling, Johan Wessberg, and Rochelle Ackerley

DOI: 10.1113/JP282873

Corresponding author(s): Roger Watkins (roger.watkins@univ-amu.fr)

Review Timeline:

Submission Date:	14-Feb-2022
Editorial Decision:	09-Mar-2022
Revision Received:	30-Mar-2022
Editorial Decision:	13-Apr-2022
Revision Received:	29-Apr-2022
Accepted:	03-May-2022

Senior Editor: Richard Carson

Reviewing Editor: Vaughan Macefield

Transaction Report:

Dear Dr Watkins,

Re: JP-RP-2022-282873 "Slowly-adapting type II afferents contribute to conscious touch sensation in humans: evidence from single unit intraneural microstimulation" by Roger Holmes Watkins, Mário Durão De Carvalho Amante, Helena Backlund Wasling, Johan Wessberg, and Rochelle Ackerley

Thank you for submitting your manuscript to The Journal of Physiology. It has been assessed by a Reviewing Editor and by 2 expert Referees and I am pleased to tell you that it is considered to be acceptable for publication following satisfactory revision.

The reports are copied at the end of this email. Please address all of the points and incorporate all requested revisions, or explain in your Response to Referees why a change has not been made.

NEW POLICY: In order to improve the transparency of its peer review process The Journal of Physiology publishes online as supporting information the peer review history of all articles accepted for publication. Readers will have access to decision letters, including all Editors' comments and referee reports, for each version of the manuscript and any author responses to peer review comments. Referees can decide whether or not they wish to be named on the peer review history document.

Authors are asked to use The Journal's premium BioRender (<https://biorender.com/>) account to create/redraw their Abstract Figures. Information on how to access The Journal's premium BioRender account is here: <https://physoc.onlinelibrary.wiley.com/journal/14697793/biorender-access> and authors are expected to use this service. This will enable Authors to download high-resolution versions of their figures. The link provided should only be used for the purposes of this submission. Authors will be charged for figures created on this premium BioRender account if they are not related to this manuscript submission.

I hope you will find the comments helpful and have no difficulty returning your revisions within 4 weeks.

Your revised manuscript should be submitted online using the links in Author Tasks Link Not Available.

Any image files uploaded with the previous version are retained on the system. Please ensure you replace or remove all files that have been revised.

REVISION CHECKLIST:

- Article file, including any tables and figure legends, must be in an editable format (eg Word)
- Abstract figure file (see above)
- Statistical Summary Document
- Upload each figure as a separate high quality file
- Upload a full Response to Referees, including a response to any Senior and Reviewing Editor Comments;
- Upload a copy of the manuscript with the changes highlighted.

- A potential 'Cover Art' file for consideration as the Issue's cover image;
- Appropriate Supporting Information (Video, audio or data set https://jp.msubmit.net/cgi-bin/main.plex?form_type=display_requirements#supp).

To create your 'Response to Referees' copy all the reports, including any comments from the Senior and Reviewing Editors, into a Word, or similar, file and respond to each point in colour or CAPITALS and upload this when you submit your revision.

I look forward to receiving your revised submission.

If you have any queries please reply to this email and staff will be happy to assist.

Yours sincerely,

Richard Carson
Senior Editor
The Journal of Physiology

REQUIRED ITEMS:

-Author photo and profile. First (or joint first) authors are asked to provide a short biography (no more than 100 words for one author or 150 words in total for joint first authors) and a portrait photograph. These should be uploaded and clearly labelled with the revised version of the manuscript. See Information for Authors for further details.

-You must start the Methods section with a paragraph headed Ethical Approval. If experiments were conducted on humans confirmation that informed consent was obtained, preferably in writing, that the studies conformed to the standards set by the latest revision of the Declaration of Helsinki, and that the procedures were approved by a properly constituted ethics committee, which should be named, must be included in the article file. If the research study was registered (clause 35 of the Declaration of Helsinki) the registration database should be indicated, otherwise the lack of registration should be noted as an exception (e.g. The study conformed to the standards set by the Declaration of Helsinki, except for registration in a database.). For further information see: <https://physoc.onlinelibrary.wiley.com/hub/human-experiments>

-Please upload separate high-quality figure files via the submission form.

-A Statistical Summary Document, summarising the statistics presented in the manuscript, is required upon revision. It must be on the Journal's template, which can be downloaded from the link in the Statistical Summary Document section here: https://jp.msubmit.net/cgi-bin/main.plex?form_type=display_requirements#statistics

-Please include an Abstract Figure. The Abstract Figure is a piece of artwork designed to give readers an immediate understanding of the research and should summarise the main conclusions. If possible, the image should be easily 'readable' from left to right or top to bottom. It should show the physiological relevance of the manuscript so readers can assess the importance and content of its findings. Abstract Figures should not merely recapitulate other figures in the manuscript. Please try to keep the diagram as simple as possible and without superfluous information that may distract from the main conclusion(s). Abstract Figures must be provided by authors no later than the revised manuscript stage and should be uploaded as a separate file during online submission labelled as File Type 'Abstract Figure'. Please ensure that you include the figure legend in the main article file. All Abstract Figures should be created using BioRender. Authors should use The Journal's premium BioRender account to export high-resolution images. Details on how to use and access the premium account are included as part of this email.

EDITOR COMMENTS

Reviewing Editor:

I have now received the reports from two independent reviewers, both experts in the field of human tactile neurophysiology. I am pleased to report that your manuscript is considered potentially acceptable, subject to your satisfactory revision according to the reviewer's comments.

Approval numbers need to be provided from the three ethics committees overseeing these studies

REFEREE COMMENTS

Referee #1:

By using microneurography and microstimulation of single tactile afferents in human subjects this study aims to define the perceived tactile sensations evoked from SA-II afferents. This is an interesting scientific question since ambiguous results have been presented, and the general perception today is that SA-II afferents do not present a robust percept upon

microstimulation.

This is a well written manuscript that I find very interesting. The method is sound, the results are convincingly presented, and the discussion is appropriate. I just have a few minor comments:

1. In Fig. 1 B the graph showing the firing rate of the afferent ends to early. Correct the figure.

2. Microstimulation of single afferents is dependent on a small tip of the electrodes. What type of electrodes were used in the different experiments and what was their impedance? Was the impedance measured in situ?

Referee #2:

This manuscript addresses a long-debated question about whether activation of a single SA-II afferent in the glabrous skin can evoke conscious percept. This is the first study systematically investigating this question and presenting compelling evidence that this might be the case. It uses highly sophisticated intraneural microstimulation (INMS) techniques via microneurography electrodes in humans. This technique is currently available in only a couple of laboratories in the world and only a handful of studies have ever been published using it.

It is concluded that activation of SA-II afferents does indeed evoke a characteristic tactile sensation distinct from other afferents. The evidence seems to be as strong as that for other afferent types derived using the INMS technique. Despite the discrepancy of this finding with some previous studies, the result could be anticipated from some earlier work. For example, the study of Olausson et al 1998 demonstrated that 0.13 mm lateral skin stretch is sufficient to evoke clear perception and identification of pull direction on the skin surface is attributed specifically to SA-II afferents. Thus there is no doubt that SA-IIs in the hairy skin strongly contribute to perception. There is no reasonable argument why this wouldn't apply to glabrous skin. However it is not known whether more than one afferent would be required to be activated at the same time. One of the reasons why doubts about the ability of SA-IIs to contribute to perception have been reinforced and maintained is that monkeys don't have them in glabrous skin and thus human perception models based on electrophysiological data recorded in monkeys have no choice but disregard them and argue about their limited contribution to perception.

The data obtained in this study are of exceptional interest and importance; however the manuscript needs some improvements.

Introduction

Page 3: "In the human hand" - should be glabrous skin as hairy skin is different.

"(vibration, fine texture) are conveyed by the fast conducting afferents" - these all are Ab afferents, you mean fast adapting afferents

Page 5: You have to explain what do you mean with "physiologically recorded SA-IIs". Is there also a non-physiological way to record? "their lower physiological incidence" doesn't sound right either. This is very confusing throughout the manuscript. Perceptive field is physiological too.

Material and Methods

Please report what type of amplifier and microstimulator was used. What are the technical specs of the equipment (e.g., range, accuracy) in each of the three experimental series performed in two different locations using different setups. Please give specs of charge balanced pulse shape. Full manufacturer's specs of electrodes should be given as the authors themselves indicate that this might be a crucial factor explaining some discrepancy with other studies.

One of the SA-II classification criteria is that the RFs in all SA-IIs reported here were >7mm. This doesn't appear to be the case in Figure 1B.

How was the coefficient of variation estimated? Over what time period? How were the non-spontaneous afferents stimulated

during the coefficient of variation assessment period? If afferent discharge rate over the period of time slowly drifts due to viscoelastic properties of the skin, would that contribute to the value of the coefficient of variation?

Results

Was there a risk of experimental bias that afferents for which electrophysiological type didn't match the expected perceptual properties were rejected? Apart from RF location mismatch (which wouldn't happen very often due to large size of SA-II RFs and distinct well ordered somatotopic fascicle organization at the wrist level), did you use RF size mismatch and type of sensation it evokes to reject units as a mismatch?

For example:

- 1) If an SA-II afferent is recorded from, but the perceptive field size is small and sensation sharp (pinching pressure), while its location easily falls within the large RF of the recorded SA-II, would it be rejected?
- 2) If an SA-II afferent is recorded from, but the perceptive field size is small and sensation sharp (vibration, strobing, tingling), would it be rejected?
- 3) If an SA-I afferent is recorded from, but its perceptive field size is large with diffuse borders, will it be rejected as a mismatch?

Discussion

Page 17, top paragraph. One has to acknowledge that even a rough 4 channel cuff electrode wrapped around the nerve stimulating a bundle of multiple afferents can evoke very distinct natural clear sensations like pressure (Tan et al 2014), and not just electrical shock/paresthesia. Thus it seems that naturalistic sensation is not a criterion to determine whether one single afferent is stimulated.

Page 18, top paragraph. The reasoning around aberrant firing and questioning successful afferent recruitment in Ocha and Torebjörk study is not clear. This section has to be rewritten clarifying the technique used and better explaining the shortcomings which are suggested might invalidate their conclusions.

Page 18 bottom paragraph and Page 19 top. The relevance of electrical pulse width in stimulation of single afferents is not clear. The discussion here on INMS stimulation parameters with reference to the Tan et al 2014 study changing sensation quality using cuff electrodes confuses single afferent stimulation techniques with multiunit nerve stimulation. Unlike whole nerve stimulation, one afferent responds in an all-or-nothing fashion, therefore changing supra-threshold stimulus parameters while stimulating a single afferent would have no effect on perceived quality.

It is also not clear why SA-IIs might have been influenced by stimulation parameters while other afferent types wouldn't? Have the authors found evidence that electrical excitability properties of SA-II afferents innervating Ruffini endings is different from other afferent types and thus they would require different stimuli to be activated?

Page 19, second paragraph. Incorrect references. Johansson & Birznieks did not study microslips or safety margins, instead you should refer to studies which specifically investigated how safety margin and partial slips are signaled by tactile afferents:

DOI: 10.1109/EMBC.2014.6944531 Khamis et al 2004: Tactile afferents encode grip safety before slip for different frictions.

DOI: 10.7554/eLife.64679 Delhaye et al 2021: High-resolution imaging of skin deformation shows that afferents from human fingertips signal slip onset.

END OF COMMENTS

Confidential Review

14-Feb-2022

JP-RP-2022-282873: Slowly-adapting type II afferents contribute to conscious touch sensation in humans: evidence from single unit intraneural microstimulation

EDITOR COMMENTS

Reviewing Editor: I have now received the reports from two independent reviewers, both experts in the field of human tactile neurophysiology. I am pleased to report that your manuscript is considered potentially acceptable, subject to your satisfactory revision according to the reviewer's comments. Approval numbers need to be provided from the three ethics committees overseeing these studies.

We thank the editor for their input and we have added the approval numbers from the ethical committees overseeing these studies. As required by the Journal of Physiology, we also start the Methods section with a paragraph headed 'Ethical Approval' (page 6).

REFEREE COMMENTS

Referee #1: By using microneurography and microstimulation of single tactile afferents in human subjects this study aims to define the perceived tactile sensations evoked from SA-II afferents. This is an interesting scientific question since ambiguous results have been presented, and the general perception today is that SA-II afferents do not present a robust percept upon microstimulation. This is a well written manuscript that I find very interesting. The method is sound, the results are convincingly presented, and the discussion is appropriate. I just have a few minor comments:

We thank the reviewer for their positive responses on our manuscript and the constructive suggestions for improvement.

1. In Fig. 1 B the graph showing the firing rate of the afferent ends too early. Correct the figure.

We thank the reviewer for spotting this and the error with the x-axis scaling of the firing rate trace has been corrected.

2. Microstimulation of single afferents is dependent on a small tip of the electrodes. What type of electrodes were used in the different experiments and what was their impedance? Was the impedance measured in situ?

To provide clarity on this point we have added details to the methods section on electrode characteristics and impedance measurements (bottom of page 7).

"Commercial tungsten microelectrodes (FHC, Bowdoin, ME; model UNA15FNM) with tip diameters of ~5 μm were inserted perpendicularly into the skin and adjusted to an intraneural position for neural recordings."

"Using these electrodes, matched recording and stimulation in single afferents (see below) was achieved with electrodes having impedances typically measured as between 150-500 k Ω in situ at 1 kHz (Glover et al., 2017)."

Referee #2: This manuscript addresses a long-debated question about whether activation of a single SA-II afferent in the glabrous skin can evoke conscious percept. This is the first study systematically investigating this question and presenting compelling evidence that this might be the case. It uses highly sophisticated intraneural microstimulation (INMS) techniques via microneurography electrodes in humans. This technique is currently available in only a couple of laboratories in the world and only a handful of studies have ever been published using it. It is concluded that activation of SA-II afferents does indeed evoke a characteristic tactile sensation distinct from other afferents. The evidence seems to be as strong as that for other afferent types derived using the INMS technique. Despite the discrepancy of this finding with some previous studies, the result could be anticipated from some earlier work. For example, the study of Olausson et al 1998 demonstrated that 0.13 mm lateral skin stretch is sufficient to evoke clear perception and identification of pull direction on the skin surface is attributed specifically to SA-II afferents. Thus there is no doubt that SA-IIs in the hairy skin strongly contribute to perception. There is no reasonable argument why this wouldn't apply to glabrous skin. However it is not known whether more than one afferent would be required to be activated at the same time. One of the reasons why doubts about the ability of SA-IIs to contribute to perception have been reinforced and maintained is that monkeys don't have them in glabrous skin and thus human perception models based on electrophysiological data recorded in monkeys have no choice but disregard them and argue about their limited contribution to perception. The data obtained in this study are of exceptional interest and importance; however the manuscript needs some improvements.

We thank the reviewer for their highly insightful comments and suggestions on our manuscript. The reviewer makes a very good point about the contributions of hairy skin SA-IIs to sensation. The Olausson et al (1998) reference has been added to the section mentioning hairy skin SA-IIs in the discussion (middle of page 22). This is definitely a promising area for future work, but since substantial differences exist between the SA-II afferents in the hairy and glabrous skin that may produce differences in response properties and conveyed tactile sensations (receptive field size, force activation threshold, propensity for spontaneous activity), this is a relatively complex point and we have kept this discussion brief. This will be an interesting question to assess, additionally in the context that even less is known about individual afferent linked precepts in hairy skin, with data limited to few examples (Nagi et al., 2019; Schady and Torebjork 1983), with no direct mention of attempting SA-II hairy skin stimulation.

Introduction

Page 3: "In the human hand" - should be glabrous skin as hairy skin is different. "(vibration, fine texture) are conveyed by the fast conducting afferents" - these all are Ab afferents, you mean fast adapting afferents

Thank you for spotting these. Both errors have been corrected in the text

Page 5: You have to explain what do you mean with "physiologically recorded SA-IIs". Is there also a non-physiological way to record? "their lower physiological incidence" doesn't sound right either. This is very confusing throughout the manuscript. Perceptive field is physiological too.

We agree this is not specific enough and have replaced the physiological phrase at several points in the manuscript, instead referring to these as individual characterized afferents.

Material and Methods

Please report what type of amplifier and microstimulator was used. What are the technical specs of the equipment (e.g., range, accuracy) in each of the three experimental series performed in two different locations using different setups. Please give specs of charge balanced pulse shape. Full manufacturer's specs of electrodes should be given as the authors themselves indicate that this might be a crucial factor explaining some discrepancy with other studies.

We have added technical details to the methods giving the resolution ($\leq 0.1 \mu\text{A}$) and range (up to $10 \mu\text{A}$) of the equipment used for the microstimulation (from the bottom of page 7):

“Electrodes were connected to custom designed amplifiers (see REF Glover 2017) capable of amplifying and bandpass filtering nerve activity ($\sim 0.2\text{-}4\text{kHz}$) for display and saving. Experimental series A used the SC/Zoom setup (University of Umeå, Umeå, Sweden) and experimental series B and C used the microneurography recording/stimulation setup detailed in Glover et al. (2017). Both systems had integrated nerve stimulators, which were capable of delivering stimulation at intensities of up to $\sim 200 \mu\text{A}$ with a $\leq 0.1 \mu\text{A}$ resolution. Up to $10 \mu\text{A}$ stimulation used in the present experiments delivering intraneural stimulation through high impedance electrodes.”

We have also added more information regarding the charge balanced pulses used (middle of page 10):

“In series B and C the pulses were followed by a full active charge-balancing, using a 2 ms negative pulse (Glover et al., 2017).”

Full specifications of the electrodes have been provided, along with a brief description of the impedance measurement made in situ (mid-bottom of page 7/top of page 8):

“Using these electrodes, unitary matched recording and stimulation (see below) was achieved with electrodes having impedances measured typically between $150\text{-}500 \text{K}\Omega$ in situ at 1kHz (Glover et al., 2017).”

One of the SA-II classification criteria is that the RFs in all SA-IIs reported here were $>7\text{mm}$. This doesn't appear to be the case in Figure 1B. How was the coefficient of variation estimated? Over what time period? How were the non-spontaneous afferents stimulated during the coefficient of variation assessment period? If afferent discharge rate over the period of time slowly drifts due to viscoelastic properties of the skin, would that contribute to the value of the coefficient of variation?

We understand that the representation of the receptive fields on Figure 1B has produced confusion, as the point only indicated location, not receptive field size. The legend explains that it is the receptive field location that is shown in the figure, but we have now also labelled it on the illustration for clarity. For the recording characterisation, we feel that the location of the receptive field center is perhaps better to illustrate, as this is where the focal mechanical stimulation test for matching sensations was conducted in all afferents (more detail has also added on this point below). For experiment series B and C, receptive fields were systematically mapped, and we realize this was somewhat ambiguously stated in the methods ‘all measured SA-IIs’. We have made this point more specifically, that the systematic mapping and measurements are based on series’ B and C where an example of precise correspondence between receptive and perceptive fields can be demonstrated (Figure 2B).

We provide information about receptive field size, but we do not specifically include it in the criteria stated in the middle of page 9. Although this mapping is helpful in distinguishing between glabrous SA-Is and SA-IIs, it is not necessarily true for hairy skin units that all have small receptive fields (Vallbo et al., 1995). Receptive field size mapping is also dependent on the stimulus used. The receptive field measurement is associated with afferent type, given for reference and has been kept in the text. However, a coefficient of variation measurement in combination with the manual tests is fully sufficient for accurate classification, even in skin areas where the field size difference is not present between SA-Is and SA-IIs (Chambers et al., 1972).

The coefficient of variation was estimated by analysing the standard deviation of the interspike intervals during the static phase of firing to controlled sustained monofilament indentation. This has been detailed at the bottom of page 9, where we take at least 100 datapoints after the initial burst response on monofilament application, as in the literature (Chambers et al., 1972; Knibestöl, 1975; Gynther et al., 1992). In our sample, 2/18 SA-IIs were spontaneously active (stated at the beginning of the Results), but this spontaneous activity can depend upon the skin tension. As typically done, we only measured the coefficient of variation to sustained indentation, it was not calculated based on spontaneous firing. Although there could be slow drifts in discharge rate due to viscoelastic properties of the skin, we do not believe that would change the coefficient of variation much, as we measured it over seconds, not minutes. Further, this could mean a slight increase in the coefficient of variation, but our SA-II values were all low and under 0.3.

Results

Was there a risk of experimental bias that afferents for which electrophysiological type didn't match the expected perceptual properties were rejected? Apart from RF location mismatch (which wouldn't happen very often due to large size of SA-II RFs and distinct well ordered somatotopic fascicle organization at the wrist level), did you use RF size mismatch and type of sensation it evokes to reject units as a mismatch?

The reviewer raises a very good point about the confidence in matching of the receptive fields. We do not believe there would have been a high experimental bias for this, as we were careful with the procedure. Although we did likely consider the receptive field size, this is a more ambiguous measure, both in the receptive field mapping (e.g. the size can be dependent on the mechanical stimulation used) and in the perceptive field sensation (e.g. diffuse borders can make it more difficult to define the sensation, which can vary depending on individual perception). Further, previous work uses criteria of overlapping receptive fields and type of sensation (Vallbo et al. 1984; Torebjörk et al. 1987). To be clearer and elaborate on the propensity for misidentifying non-matching sensations, we have made the specifics of the mechanical matching test clearer (page 11)

“The correspondence between the receptive field of the individual identified afferent and the INMS evoked perceptive field was verified using alternating mechanical and electrical stimulation. Localized mechanical stimulation was delivered using indentation with a blunt wooden stick (tip diameter ~1 mm) at the identified point of maximal sensitivity alternately with the pulse trains used for initial identification of sensations. Stimulation was only continued if; 1) there was a very close correspondence between the physiological receptive field and the INMS evoked receptive field, with exactly overlapping percepts or located within a few mm (Vallbo et al. 1984; Torebjörk et al. 1987) (Figure 2B) and 2) if the quality of the sensation (cyclic vs.

sustained) matched the adaptation characteristics of the afferent (fast vs slow adaptation, respectively). This careful evaluation of correspondence between stimulation and recording...”

Identifying sensation overlap at the maximal point of sensitivity protects against afferents with large receptive fields overlapping with unrelated focal precepts. Even though there is general somatotopic organisation of the median nerve at the wrist level, when it comes to INMS, at a single electrode site INMS percept locations evoked by changing stimulus intensity alone may differ significantly in location, in fact spanning most of the median nerve territory (see Fig 3, Vallbo et al., 1984). Thus, unrelated precepts may appear somewhat randomly in this region, reducing the likelihood of overlaps being observed by chance, which matches our observations on unrelated sensations which are recruited when increasing intensity so that sensation intensity plateaus.

For example: 1) If an SA-II afferent is recorded from, but the perceptive field size is small and sensation sharp (pinching pressure), while its location easily falls within the large RF of the recorded SA-II, would it be rejected?

In this example, the sensation would potentially be considered as matching if there was a precise correspondence between the SA-II sensitive zone (which is in fact usually a relatively focal point; Johansson 1976) and the point of sensation, but with the possibility that this could be a chance overlapping mismatch with an SA-I afferent. Considering the relatively large area of skin innervated by even a single fascicle spanning tens of millimeters (Hallin 1990), the likelihood of such a precise overlap with this identification strategy is fairly minimal and would not be seen consistently at a population level as the SA-II sensations were.

2) If an SA-II afferent is recorded from, but the perceptive field size is small and sensation sharp (vibration, strobing, tingling), would it be rejected?

For this instance, the sensation would be rejected on the basis that the quality of the sensation (e.g. sustained vs cyclic) did not match the afferent type (now elaborated in the methods, page 11),

“2) if the quality of the sensation (cyclic vs. sustained) matched the adaptation characteristics of the afferent (fast vs slow adaptation, respectively).”

since this has been consistently reported for the matching precepts from FA-I, SA-I and FA-II afferents.

3) If an SA-I afferent is recorded from, but its perceptive field size is large with diffuse borders, will it be rejected as a mismatch?

This is a more difficult case, since focal stimulation the receptive field center method does not account for a larger potentially overlapping sensation area. There is a strong precedent for the type of sensation associated with SA-I afferents, both in terms of the size of sensation (very focal) and the perceptual descriptors (sharp/pinch/focal), so this would represent an unusual case. Only a well centered sensation of a such a quality when recording from an SA-I would be considered for acceptance as a potentially matching sensation and would either be rejected or marked as a clear outlier in terms of sensation size/quality. Again, on theoretical grounds this is still not a particularly likely situation, the first sensation matches the location of a well isolated single afferent in 50% of cases, and even the maximal sensation size of broad pressure (being less than the region of mass activity in a fascicle; Schady et al., 1983, Hallin 1990), the often large differences in location of separate sensations recruited at a single electrode position

(Vallbo 1984) and the fact these are observed in around 10% of our first INMS sensations sample leads such a case of overlap to be relatively unlikely.

Discussion

Page 17, top paragraph. One has to acknowledge that even a rough 4 channel cuff electrode wrapped around the nerve stimulating a bundle of multiple afferents can evoke very distinct natural clear sensations like pressure (Tan et al 2014), and not just electrical shock/paresthesia. Thus it seems that naturalistic sensation is not a criterion to determine whether one single afferent is stimulated.

We thank the reviewer for raising this and we do agree that in special cases of stimulation (such as the modulation used by Tan et al.) can produce more naturalistic percepts. In the Tan et al. paper, they clearly state that constant stimulation intensity produced paresthesia, whereas when they modulated the stimulation using time-variant pulses, the sensation of pressure was felt. This is quite different to our approach, where we have a very controlled input; however, in the modulated cuff approach, it is difficult to know what is being stimulated, which could generate spikes in various afferents at various times, thus contributing to a different stimulus-evoked percept. In either case, it is clear that the electrical stimulation of many afferents simultaneously produces electrical shock/paresthesia, there does not seem to be any doubt in that.

Therefore, we do not say that naturalistic sensation is a criterion in general for determining whether one afferent is stimulated, but in the case of our stimulation, this is still valid. Thus, we have modified this part of the discussion (top of page 19) to be more specific. We readdress the findings of Tan et al. further in modifications at the bottom of page 20.

Page 18, top paragraph. The reasoning around aberrant firing and questioning successful afferent recruitment in Ochoa and Torebjörk study is not clear. This section has to be rewritten clarifying the technique used and better explaining the shortcomings which are suggested might invalidate their conclusions.

We agree that this discussion was confusing and perhaps not necessary, so have only mentioned this in brief as an indirect technique that has been used to indicate successful recruitment without sensation. This is extremely complicated issue, but the point is that the approach from Ochoa and Torebjörk is an indirect measurement (page 19).

“An indirect method has been used to infer successful activation despite a lack of sensation (Torebjörk and Ochoa 1980), but involves prolonged high frequency stimulation which may induce significant nerve polarisation.”

Page 18 bottom paragraph and Page 19 top. The relevance of electrical pulse width in stimulation of single afferents is not clear. The discussion here on INMS stimulation parameters with reference to the Tan et al 2014 study changing sensation quality using cuff electrodes confuses single afferent stimulation techniques with multiunit nerve stimulation. Unlike whole nerve stimulation, one afferent responds in an all-or-nothing fashion, therefore changing supra-threshold stimulus parameters while stimulating a single afferent would have no effect on perceived quality. It is also not clear why SA-IIs might have been influenced by stimulation parameters while other afferent types wouldn't?

To our knowledge, the specific excitability properties of the different afferent classes have not been examined using variable electrical stimulation, but there are excitability differences between sensory and motor axons (below). When studying electrical responses in myelinated afferents to calculate conduction velocity for example, it is common to use stimulation intensities at twice threshold to prevent any variation in conduction velocity or recruitment (see Harper and Lawson 1985). As discussed above and in the methods, we are in the region below this, where there may be some uncertainty about recruitment and the recruitment may be critically dependent on excitability and subtle differences in the stimulation parameters. The Tan et al., 2014 paper is however relevant as it illustrates that the pulse width can clearly have an effect on recruitment of afferents independent of amplitude, but there has been no examination of the determinants of differential activation in this case and whether this bears any relation to the afferent type so this is somewhat speculative.

Have the authors found evidence that electrical excitability properties of SA-II afferents innervating Ruffini endings is different from other afferent types and thus they would require different stimuli to be activated?

With reference to the excitability of different fiber types, there are only a handful of studies that to our knowledge have directly examined these in humans (Bostock and Rothwell 1997; Mogyoros et al., 1996; Bostock et al., 1983), and we have added these references during the discussion of the origin of the discrepancies in the discussion. All find there are complex determinants of recruitment properties during electrical stimulation, particularly around threshold, and that these can differ between different types of peripheral axons (sensory vs motor). Thus, there is precedent for some speculation on this point to explain the discrepancies to previous work.

Page 19, second paragraph. Incorrect references. Johansson & Birznieks did not study microslips or safety margins, instead you should refer to studies which specifically investigated how safety margin and partial slips are signaled by tactile afferents: DOI: 10.1109/EMBC.2014.6944531 Khamis et al 2004: Tactile afferents encode grip safety before slip for different frictions. DOI: 10.7554/eLife.64679 Delhaye et al 2021: High-resolution imaging of skin deformation shows that afferents from human fingertips signal slip onset.

We thank the reviewer for spotting these errors and the references have been corrected

Dear Dr Watkins,

Re: JP-RP-2022-282873R1 "Slowly-adapting type II afferents contribute to conscious touch sensation in humans: evidence from single unit intraneural microstimulation" by Roger Holmes Watkins, Mário Durão De Carvalho Amante, Helena Backlund Wasling, Johan Wessberg, and Rochelle Ackerley

Thank you for submitting your manuscript to The Journal of Physiology. It has been assessed by a Reviewing Editor and by 2 expert Referees and I am pleased to tell you that it is considered to be acceptable for publication following satisfactory revision.

The reports are copied at the end of this email. Please address all of the points and incorporate all requested revisions, or explain in your Response to Referees why a change has not been made.

NEW POLICY: In order to improve the transparency of its peer review process The Journal of Physiology publishes online as supporting information the peer review history of all articles accepted for publication. Readers will have access to decision letters, including all Editors' comments and referee reports, for each version of the manuscript and any author responses to peer review comments. Referees can decide whether or not they wish to be named on the peer review history document.

Authors are asked to use The Journal's premium BioRender (<https://biorender.com/>) account to create/redraw their Abstract Figures. Information on how to access The Journal's premium BioRender account is here: <https://physoc.onlinelibrary.wiley.com/journal/14697793/biorender-access> and authors are expected to use this service. This will enable Authors to download high-resolution versions of their figures. The link provided should only be used for the purposes of this submission. Authors will be charged for figures created on this premium BioRender account if they are not related to this manuscript submission.

I hope you will find the comments helpful and have no difficulty returning your revisions within 4 weeks.

Your revised manuscript should be submitted online using the links in Author Tasks Link Not Available.

Any image files uploaded with the previous version are retained on the system. Please ensure you replace or remove all files that have been revised.

REVISION CHECKLIST:

- Article file, including any tables and figure legends, must be in an editable format (eg Word)
- Abstract figure file (see above)
- Statistical Summary Document
- Upload each figure as a separate high quality file
- Upload a full Response to Referees, including a response to any Senior and Reviewing Editor Comments;
- Upload a copy of the manuscript with the changes highlighted.

- A potential 'Cover Art' file for consideration as the Issue's cover image;
- Appropriate Supporting Information (Video, audio or data set https://jp.msubmit.net/cgi-bin/main.plex?form_type=display_requirements#supp).

To create your 'Response to Referees' copy all the reports, including any comments from the Senior and Reviewing Editors, into a Word, or similar, file and respond to each point in colour or CAPITALS and upload this when you submit your revision.

I look forward to receiving your revised submission.

If you have any queries please reply to this email and staff will be happy to assist.

Yours sincerely,

Richard Carson
Senior Editor
The Journal of Physiology

REQUIRED ITEMS:

-Papers must comply with the Statistics Policy https://jp.msubmit.net/cgi-bin/main.plex?form_type=display_requirements#statistics

In summary:

-If $n \leq 30$, all data points must be plotted in the figure in a way that reveals their range and distribution. A bar graph with data points overlaid, a box and whisker plot or a violin plot (preferably with data points included) are acceptable formats.

-If $n > 30$, then the entire raw dataset must be made available either as supporting information, or hosted on a not-for-profit repository e.g. FigShare, with access details provided in the manuscript.

- n clearly defined (e.g. x cells from y slices in z animals) in the Methods. Authors should be mindful of pseudoreplication.

-All relevant n values must be clearly stated in the main text, figures and tables, and the Statistical Summary Document (required upon revision)

-The most appropriate summary statistic (e.g. mean or median and standard deviation) must be used. Standard Error of the Mean (SEM) alone is not permitted.

-Exact p values must be stated. Authors must not use 'greater than' or 'less than'. Exact p values must be stated to three significant figures even when 'no statistical significance' is claimed.

-Statistics Summary Document completed appropriately upon revision

EDITOR COMMENTS

Reviewing Editor:

Dear Dr Watkins,

Reviewer 1 is satisfied with your amendments but Reviewer 2 raises a remaining minor issue I would like you to address, following which your manuscript will be considered acceptable. I look forward to receiving your revised manuscript shortly.

Senior Editor:

Please provide additional explanatory information in the statistical summary document. It also appears as if some text has been truncated in the present version.

REFEREE COMMENTS

Referee #1:

Congratulations! I have no further comments.

Referee #2:

I would like to thank the authors for very a very interesting discussion and details provided responding to the reviewers' questions. I am satisfied with all changes.

Minor

Coefficient of variation

Figure 1 legend says "...coefficient of variation of 0.53 in the static phase of adaptation." This is inconsistent with the point-by-point reply indicating that there is no adaptation. The instantaneous discharge rate trace in Figure 1B also shows that discharge rate slowly decreases.

I don't believe it creates any major problem, but note that to evaluate afferent discharge variability over longer time interval and/or when discharge rate may slowly drift, an irregularity index (IRI) might be more appropriate measure as it is not affected by slow discharge rate changes. It is calculated as the mean of the absolute discharge rate differences measured between every two pairs of consecutive spikes divided by the mean discharge rate (DOI: 10.1113/jphysiol.2008.151746. Birznieks et al 2008).

END OF COMMENTS

1st Confidential Review

30-Mar-2022

Response to reviewers 2. JP-RP-2022-282873: Slowly-adapting type II afferents contribute to conscious touch sensation in humans: evidence from single unit intraneural microstimulation

Senior Editor

Please provide additional explanatory information in the statistical summary document. It also appears as if some text has been truncated in the present version.

The truncation was caused by resizing the columns and has been corrected, giving the additional information.

Reviewer 2

Minor: Coefficient of variation

Figure 1 legend says "...coefficient of variation of 0.53 in the static phase of adaptation." This is inconsistent with the point-by-point reply indicating that there is no adaptation. The instantaneous discharge rate trace in Figure 1B also shows that discharge rate slowly decreases.

We thank the reviewer for their very positive comments and a highly useful discussion. We agree that the wording of this part above was not that clear or consistent with exact terminology in the literature. What we wanted to refer to was 'static phase of the response' (not adaptation). We have changed this in the text (top of p. 9, top of p. 10) and it is in line with previous work (Chambers et al., 1972; Knibestöl, 1975; Gynther et al., 1992). The 'static phase' is a relative term and has historically denoted the period after the initial dynamic response, which is what we measured presently.

I don't believe it creates any major problem, but note that to evaluate afferent discharge variability over longer time interval and/or when discharge rate may slowly drift, an irregularity index (IRI) might be more appropriate measure as it is not affected by slow discharge rate changes. It is calculated as the mean of the absolute discharge rate differences measured between every two pairs of consecutive spikes divided by the mean discharge rate (DOI: 10.1113/jphysiol.2008.151746. Birznieks et al 2008).

The referee makes an excellent point that the differentiation between variability in responses may be much better assessed by a measure less affected by slow drifts in response, as in the irregularity index method presented in Birznieks et al (2008). As far as we know, this has not been applied to slowly-adapting mechanoreceptive afferent responses. With regards to our data, we agree that the slow drift in the static phase of the response would contribute somewhat to the variation in firing rate over time; however, the major contributor is still the point-to-point variation in spike rate. Thus, as the slow drifts are rather similar across our slowly-adapting units, we believe that the coefficient of variation remains a reliable measure of discharge variability, which we can compare to previous work. We will absolutely conduct analyses of the irregularity index for future studies, where we can also present the corresponding coefficient of variation. Since we focus on the SA-II coefficient of variation in our current paper, we think that it would be too much to add now, where it would complicate the paper and distract from the main SA-II message. We currently use the coefficient of variation as a positive formal confirmation of SA-IIs, in combination with the other response characteristics, but as stated before, it is not necessarily required. However, we have revised the text on this and added a point to say that the irregularity index could be more sensitive (middle of p. 9). In our future studies, we will apply sustained indentation systematically to SA-Is and SA-IIs to provide this measure as confirmation of all slowly-adapting units, using both the coefficient of variation and irregularity index – it may even show some interesting differences between measures and afferent types, which helps us further understand the variability in human mechanoreceptive afferents.

Dear Dr Watkins,

Re: JP-RP-2022-282873R2 "Slowly-adapting type II afferents contribute to conscious touch sensation in humans: evidence from single unit intraneural microstimulation" by Roger Holmes Watkins, Mário Durão De Carvalho Amante, Helena Backlund Wasling, Johan Wessberg, and Rochelle Ackerley

I am pleased to tell you that your paper has been accepted for publication in The Journal of Physiology.

NEW POLICY: In order to improve the transparency of its peer review process The Journal of Physiology publishes online as supporting information the peer review history of all articles accepted for publication. Readers will have access to decision letters, including all Editors' comments and referee reports, for each version of the manuscript and any author responses to peer review comments. Referees can decide whether or not they wish to be named on the peer review history document.

The last Word version of the paper submitted will be used by the Production Editors to prepare your proof. When this is ready you will receive an email containing a link to Wiley's Online Proofing System. The proof should be checked and corrected as quickly as possible.

Authors should note that it is too late at this point to offer corrections prior to proofing. The accepted version will be published online, ahead of the copy edited and typeset version being made available. Major corrections at proof stage, such as changes to figures, will be referred to the Reviewing Editor for approval before they can be incorporated. Only minor changes, such as to style and consistency, should be made a proof stage. Changes that need to be made after proof stage will usually require a formal correction notice.

All queries at proof stage should be sent to TJP@wiley.com

Are you on Twitter? Once your paper is online, why not share your achievement with your followers. Please tag The Journal (@jphysiol) in any tweets and we will share your accepted paper with our 23,000+ followers!

Yours sincerely,

Richard Carson
Senior Editor
The Journal of Physiology

P.S. - You can help your research get the attention it deserves! Check out Wiley's free Promotion Guide for best-practice recommendations for promoting your work at www.wileyauthors.com/eoo/guide. And learn more about Wiley Editing Services which offers professional video, design, and writing services to create shareable video abstracts, infographics, conference posters, lay summaries, and research news stories for your research at www.wileyauthors.com/eoo/promotion.

*** IMPORTANT NOTICE ABOUT OPEN ACCESS ***

Information about Open Access policies can be found here <https://physoc.onlinelibrary.wiley.com/hub/access-policies>

To assist authors whose funding agencies mandate public access to published research findings sooner than 12 months after publication The Journal of Physiology allows authors to pay an open access (OA) fee to have their papers made freely available immediately on publication.

You will receive an email from Wiley with details on how to register or log-in to Wiley Authors Services where you will be able to place an OnlineOpen order.

You can check if your funder or institution has a Wiley Open Access Account here <https://authorservices.wiley.com/author-resources/Journal-Authors/licensing-and-open-access/open-access/author-compliance-tool.html>

Your article will be made Open Access upon publication, or as soon as payment is received.

If you wish to put your paper on an OA website such as PMC or UKPMC or your institutional repository within 12 months of publication you must pay the open access fee, which covers the cost of publication.

OnlineOpen articles are deposited in PubMed Central (PMC) and PMC mirror sites. Authors of OnlineOpen articles are

permitted to post the final, published PDF of their article on a website, institutional repository, or other free public server, immediately on publication.

Note to NIH-funded authors: The Journal of Physiology is published on PMC 12 months after publication, NIH-funded authors DO NOT NEED to pay to publish and DO NOT NEED to post their accepted papers on PMC.

EDITOR COMMENTS

Dear Dr Watkins,

Thank you for attending to these remaining issues, and I am satisfied these have been adequately addressed.

2nd Confidential Review

29-Apr-2022